# OpenReview forum: "SafeDPO: A Simple Approach to Direct Preference Optimization with Enhanced Safety"
_ICLR.cc/2025/Conference — Submitted to ICLR 2025_

### Official Review · Reviewer_Thuk · 2024-11-02

**Soundness:** 3
**Presentation:** 3
**Contribution:** 3
**Rating:** 6
**Confidence:** 3

**Summary:**

This paper introduced SafeDPO which is a variation of DPO. It implemented an additional hyperparameter and used a maximum likelihood estimation with helpfulness preferences and safety indicator to eliminate the explicit training process of reward and cost models in the safe RLHF approach. The paper has shown good results via theoretical analysis and experiments. It also addressed the practical issue of samples needed via a small modification of the algorithm.

**Strengths:**

Overall, this is a novel, well-written paper which covers both the theoretical aspect and the empirical aspect. While the modifications are fairly minor compared to the original algorithms of DPO and safe RLHF, overall it has created a new algorithm with good implication in practice and can help further address safety issues in LLMs. The paper is presented logically and well-supported by theoretical analysis and experimental results - although some of the propositions are under mild assumptions, it is acceptable within the framework of the algorithm and within the scope of the paper.

**Weaknesses:**

The paper does not have any notable weaknesses. A mild observation is that using the 2 evaluation methods (using the beaver-7b-unified cost and reward models vs GPT-4), the conclusion seems to vary a lot regarding helpfulness - although they are consistent regarding harmlessness. Furthermore, the analysis might be more comprehensive if the paper delves a bit more into the model performance/helpfulness vs safety/harmlessness within subsets/categories of the dataset (which seems to be a subset of BEAVERTAILS which has 14 potential harm categories).

**Questions:**

* Why does the definition of the harmless ratio differ between model-based evaluation and GPT4 evaluation?
* The results regarding helpfulness seems to differ a lot between these two evaluation methods, could you elaborate on why this might be the case please? (for GPT4 evaluation, safe DPO and PPO-$\lambda$ have highest helpfulness but only ranked medium in model-based evaluation)
* Do you have any further analysis of the model helpfulness vs harmlessness across the harm categories as defined by the PKU-SafeRLHF-30K dataset?

**Details Of Ethics Concerns:**

Very low risk but the paper does touch on safe and responsible AI with some examples of harmful content and using human feedback.

---

> ### Author Response · Authors · 2024-11-21
>
> Thank you for your constructive feedback and comments.
>
> ### [Definition of Harmless Ratio]
> The definition of the harmless ratio is the ratio of harmless responses to all responses.
> In the model-based evaluation, the beaver-7b-unified-cost model is trained to generate a negative cost for safe answers and a positive cost for unsafe answers.
> In contrast, during GPT-4 evaluation, we let GPT-4 generate a harmlessness score between 0 and 10, where a safe answer should score higher than 5, and an unsafe answer should score lower than 5.
> This process is also based on the prompts in Safe RLHF, and we presented the details of our prompts in Appendix C.2.2. of our paper.
>
> ### [Inconsistency in Helpfulness Evaluation]
> Please refer to our general response.
>
> ### [Further Analysis]
> In this work, we focus on deriving a DAA for safety alignment.
> For future work, we plan to extend our algorithm to handle multiple objectives, such as various binary and continuous safety measures.
> As part of this future work, we will also address the reviewer's suggestion to analyze the trade-off between model helpfulness and harmlessness across harm categories, such as animal abuse, child abuse, and others.

---

> > ### Comment · Reviewer_Thuk · 2024-11-22
> >
> > Thanks for your responses which clarified my questions. I will retain the score which is in favour of accept.

---

> > > ### Author Response · Authors · 2024-11-25
> > >
> > > Thank you for taking the time to provide such thoughtful feedback.
> > > Your positive remarks and suggestions are really helpful in improving the quality of our work.

---

### Official Review · Reviewer_8s4E · 2024-11-02

**Soundness:** 3
**Presentation:** 4
**Contribution:** 3
**Rating:** 8
**Confidence:** 3

**Summary:**

This work proposes an algorithm to align LLM in the safety constrained setting. The proposed algorithm utilizes the safety indicators in the annotated datasets; and avoids helpfulness or safety modeling.

On the testing set of SafeRLHF-30K, this works demonstrates that it outperforms previous baselines, obtaining a better trade-off between helpfulness and harmfulness.

**Strengths:**

- Comparing to previous works, this work avoids helpful or safety reward modeling. Therefore, this work is more computationally efficient.
- While previous works focus on utilizing the harmless preference annotation, this work utilizes the safety indicators. This is novel.
- In the experiment section, this work demonstrates that it denominates the SOTA method PPO-$\lambda$ in both model-based and GPT-4 based evaluation.

**Weaknesses:**

- The GPT-4 model evaluation from the original Safe-RLHF paper[1] (**Evaluation Datasets** in section 4.1). It would be good to have a consistent evaluation sets so that the comparison between PPO-$\lambda$ and the proposed method is fair and straightforward. I would recommend the authors to conduct evaluation on the dataset as in [1]; or provide an explanation for why a different dataset was chosen and how it compares to the one used in [1].

- The helpfulness score of DPO-HELPFUL is lower than the rest, which is not intuitive and different from other works like [2]. This raises another concern about whether DPO baselines are fully and appropriately trained. Some analysis or explanation of the current results would be helpful.

[1] Dai, etc. Safe RLHF: Safe Reinforcement Learning from Human Feedback.
[2] Huang, etc. One-Shot Safety Alignment for Large Language Models via Optimal Dualization.

My major concerns are about the empirical evaluation. If the concerns are resolved, I am happy to change my score.

**Questions:**

- Could you provide un-normalized model-based helpfulness and harmless scores? In this way, it would be more convenient to compare with he original results in [1].

---

> ### Author Response · Authors · 2024-11-21
>
> Thank you for your constructive feedback and comments.
>
> ### [Different Dataset vs. Safe RLHF]
> First, it is challenging to determine which dataset is used in the Safe RLHF paper, as Safe RLHF involves three stages (Beaver-v1, -v2, -v3), each utilizing different datasets.
> Moreover, the Safe RLHF dataset is not yet fully available and is still being updated.
>
> When we began this research, only a 33K dataset was available in Huggingface.
> However, the dataset has been updated with a larger version.
> For the following two reasons, we conducted our experiments using SafeRLHF-30K:
> After the update, some data entries became inconsistent. For example, certain samples (x,y1,y2) were labeled such that y1 is safer than y2, yet y1 is also marked as unsafe while y2 is marked as safe.
> We aimed to use a fixed dataset to ensure consistency and avoid issues arising from dataset updates.
> We emphasize that this decision was made to maintain experimental clarity and consistency, with full acknowledgment of the Safe RLHF authors' valuable contributions.
>
> As the reviewer provided, [2] Huang, etc. One-Shot Safety Alignment for Large Language Models via Optimal Dualization. also used the same dataset.
>
> ### [Inconsistency in Helpfulness Evaluation]
> Please refer to our general response.
>
> ### [Unnormalized Scores]
>
> | Algorithm   | Reward (Normalized)     | Reward (Unnormalized) | Cost                 |
> |-------------|--------------------------|-----------------------|----------------------|
> | SafeDPO     | 4.3809 (+- 0.1099)      | 1.0390 (+- 0.0632)    | -6.2285 (+- 0.1308)  |
> | Safe RLHF   | 3.2471 (+- 0.5379)      | 0.3871 (+- 0.3093)    | -2.6617 (+- 0.6323)  |
>
> We additionally evaluated the performance of SafeDPO and Safe RLHF using three random seeds, reporting the average and standard error. Here, unnormalized rewards refer to the original rewards obtained by the reward model beaver-7b-unified-reward.
> In the revised paper, we have also reported unnormalized scores in the Appendix B.6.4.

---

> > ### Comment · Reviewer_8s4E · 2024-11-21
> >
> > Thanks for your responses. My questions have been addressed. I am happy to raise my score to 'Accept'.

---

> > > ### Author Response · Authors · 2024-11-25
> > >
> > > Thank you for taking the time to provide such thoughtful feedback.
> > > Your positive remarks and suggestions are really helpful in improving the quality of our work.

---

### Official Review · Reviewer_pSK9 · 2024-11-03

**Soundness:** 3
**Presentation:** 3
**Contribution:** 3
**Rating:** 6
**Confidence:** 2

**Summary:**

This paper tries to address an important problem regarding the safety genearation of Large Language Model. This paper proposed a novel alignment method called SafeDPO that is stable and efficient. The paper also demenstrates theotical analysis to show the safety alignment objective can be achieved with a single optimization objective. The experiments conducted by the authors show the effectiveness of the proposed SafeDPO by outperforming current baselines.

**Strengths:**

1. This work introduces a novel approach by proving that the LLM safety alignment objective can be achieved with a single optimization objective under mild assumptions. The rigorous and sound proof provided in the Appendix further validates this approach.
2. This work only requires a single-stage training compared to previous methods that might need training during muliple stages such as training reward and cost model or iteratively optimizing the objective.
3. I think the offset $\Delta$ that added to the objective function is novel.

**Weaknesses:**

1. Although the authors state that their method is efficient in times, sample usage and memory usage, I did not find any comparsion between existing methods and proposed SafeDPO to support this argument. It would help to include any training time comparisons, memory usage measurements, or sample efficiency metrics compared to baseline methods.
2. The experient results only report average performance in Figure 2, could the author include std here to show each method's stableness? For example, adding error bars to FIgure 2.
3. Minor Issue: I think the related work section should move ahead.

**Questions:**

1. As metioned in Weakness 2. Could the author give both avg and std statistics of their results?
2. From my understanding, the authors prevents the LLM generating unsafe responses by setting the reward to -$\infty$. Could this also be achieved by Multi-objective DPO[1] that has two objectives for maxmizing reward $r(x,y)$ and let the cost $c(x,y)$ less than 0. Ideally, can the author compare their methods with [1]. I understand due to the limited time, it might not feasible to train a new model to compare but can the author offers some insights and compare Multi-objective DPO to the proposed method such as a theoretical comparison or discussion of the similarities and differences between SafeDPO and Multi-objective DPO

[1] Beyond One-Preference-Fits-All Alignment: Multi-Objective Direct Preference Optimization. (https://arxiv.org/abs/2310.03708)

---

> ### Author Response · Authors · 2024-11-21
>
> Thank you for your constructive feedback and comments.
>
> ### [Efficiency of SafeDPO]
> **(Related to Q1, Weakness 1)**
>
> Please refer to our general response.
>
> ### [Stability of Algorithms]
> **(Related to Weakness 2)**
>
> | Algorithm   | Reward (Normalized)     | Reward (Unnormalized) | Cost                 |
> |-------------|--------------------------|-----------------------|----------------------|
> | SafeDPO     | 4.3809 (+- 0.1099)      | 1.0390 (+- 0.0632)    | -6.2285 (+- 0.1308)  |
> | Safe RLHF   | 3.2471 (+- 0.5379)      | 0.3871 (+- 0.3093)    | -2.6617 (+- 0.6323)  |
>
> We additionally evaluated the performance of SafeDPO and Safe RLHF using three random seeds, reporting the average and standard error. Here, unnormalized rewards refer to the original rewards obtained by the reward model beaver-7b-unified-reward.
> In addition, we have incorporated this result into Appendix B.6.4 of the revised version.
>
> ### [Comparison with MODPO]
> **(Related to Q2)**
>
> First, define reward functions similar to the SafeDPO setting.
> Let $r\_1$ be the reward function for the helpful preference, $r\_2$ be the reward function for safety, and $\mathbf{w}=(1,1)$.
> Then, we can obtain the following MODPO objective for $k=1$:
> \begin{equation*}
> -\mathbb{E}\_{\mathcal{D}\_1}\bigg[\log\sigma\bigg(\beta\log\frac{\pi_\theta(y\_w\vert x)}{\pi_\text{ref}(y\_w\vert x)}-\beta\log\frac{\pi_\theta(y\_l\vert x)}{\pi_\text{ref}(y\_l\vert x)}-m(x,y\_w,y\_l)\bigg)\bigg],
> \end{equation*}
> where
> \begin{equation*}
> m(x,y\_w,y\_l)=r\_2(x,y\_w)-r\_2(x,y\_l)\in\lbrace-\infty,0,\infty\rbrace.
> \end{equation*}
> Here, the differences between the SafeDPO objective and this modified MODPO objective are as follows:
>
> 1. Additional terms: $-(\tilde h\_l - \tilde h\_w)\Delta$ for SafeDPO and $-m(x,y\_w,y\_l)$ for MODPO.
> 2. Reordering preference: In SafeDPO, if $y\_l$ is safe and $y\_w$ is unsafe, the preference is reordered as $\tilde y\_w=y\_l$ and $\tilde y\_l=y\_w$. In contrast, MODPO does not consider reordering preferences.
>
> From 1, consider the following case:
> If we set
> \begin{equation*}
> r\_2(x,y)=\begin{cases}0 & \text{if $y$ is safe} \\\\ -\infty & \text{otherwise}\end{cases},
> \end{equation*}
> then following challenging arises: when $m=\infty$ (i.e., $y\_w$ is safe and $y\_l$ is unsafe) or $m=-\infty$ (i.e., $y\_w$ is unsafe and $y\_l$ is safe), it becomes impossible to minimize the objective since the gradient signal becomes zero or $\pm\infty$.
> To address this limitation, we can instead set
> \begin{equation*}
> r\_2(x,y)=\begin{cases}0 & \text{if $y$ is safe} \\\\ -\Delta & \text{otherwise}\end{cases},
> \end{equation*}
> then $m(x,y\_w,y\_l)=(-h\_w + h\_l)\Delta$, which avoids the problem of infinite values and aligns more closely with the SafeDPO objective.
> However, another issue still arises due to the difference (2): even if $y\_l$ is safe and $y\_w$ is unsafe, the objective attempts to increase the probability of $y\_w$, which is unsafe, and decrease the probability of $y\_l$, which is safe.
> Therefore, although a similar objective to SafeDPO can be derived from the MODPO objective, a simple modification of MODPO is insufficient to properly address the safety alignment objective.
>
> Consequently, MODPO is not well-suited for handling binary variables, whereas SafeDPO cannot address continuous safety variables.
> As the reviewer pointed out, both MODPO and SafeDPO can be utilized to enhance the safety of LLMs by leveraging continuous and binary safety variables, respectively.
> For future work, we aim to extend our algorithm to address multiple objectives, incorporating both binary and continuous safety measures.
> We believe that combining MODPO and SafeDPO represents one of the most promising directions to achieve this goal.
>
> ### [Related work]
> In the revised version, we have moved the related work section to precede the main section, shifting it from Section 4 to Section 3.

---

> > ### Comment · Reviewer_pSK9 · 2024-11-21
> >
> > Thanks for the detailed response! I am still in favor of accepting this paper.

---

> ### Author Response · Authors · 2024-11-25
>
> Thank you for taking the time to provide such thoughtful feedback.
> Your positive remarks and suggestions are really helpful in improving the quality of our work.

---

### Official Review · Reviewer_WZ5n · 2024-11-04

**Soundness:** 3
**Presentation:** 2
**Contribution:** 3
**Rating:** 6
**Confidence:** 4

**Summary:**

This paper proposes SafeDPO, a safety alignment fine-tuning algorithm for LLMs that aims to reduce time and memory overhead from SafeRLHF. RLHF fine-tuning uses a pair of human-labeled (more preferred vs. less preferred) outputs to fine-tune LLM. It is complex because it takes extra efforts to fit a reward and a cost model, while DAA reduces complexity by fitting the LLM policy end-to-end. SotA RLHF algorithms have already been used in solving the safety alignment problem (SafeRLHF), and the goal of SafeDPO is to find a DAA version of solution. The trick here is to set reward to negative infinity for unsafe responses (cost < 0) as in Equation 10. On top of this reward definition, the authors show a way to construct the supervised learning dataset, by swapping the more preferred and less preferred LLM outputs based if the more preferred one is considered more harmful. Furthermore, an additional offset into the loss function is introduced, as a multiplier to the difference between the harmfulness of the two outputs.

**Strengths:**

1. The motivation is good: to reduce computational complexity in RLHF-based safety alignment.

**Weaknesses:**

1. The contribution seems incremental.
(1) Section 3.1 only provides a piecewise reward function, by setting all harmful responses with negative infinity reward.
(2) Section 3.2 provides a swap of more preferred vs. less preferred labels based on harmfulness to preprocess training data, which confuses me (see my point 3 below).
(3) Section 3.3 adds an additional fine-tuning hyperparameter \Delta, but the effect is trivial. As shown in Fig 4, large \Delta has approximately the same harmless ratio as no \Delta (\Delta = 0), and helpfulness even degenerates as \Delta increases. Introducing this hyperparameter seems to have a negative effect according to the experiment results.

2. In Equation 10, where does r(x, y) and c(x, y) come from? Are they obtained the same way as in other DAA methods, such as DPO?

3. In Proposition 3.2, the term "... can be regarded as sampled from ..." is a confusing expression. I do not get what does it mean.

4. The major contribution claimed by this paper is the computational complexity reduction from RLHF baselines, but there is no experimental complexity comparison or analysis.

5. The presentation is hard to follow because too many notations are used.

**Questions:**

Please see weaknesses. If they are addressed properly, I am willing to raise my score.

---

> ### Author Response · Authors · 2024-11-21
>
> Thank you for your constructive feedback and comments.
>
> ### [Contribution]
> **(Related to Weakness 1)**
>
> First, we apologize for the potential impact on readability caused by the use of many notations.
> We acknowledge that the use of multiple notations can make the paper harder to follow.
> We are actively considering ways to improve readability while preserving the formal derivations of our formulations, and we will address this in a future revised version.
>
> (1), (2), please refer to our general response.
>
> (3) The core idea of SafeDPO is reordering: (safe and helpful response) > (safe but unhelpful response) >> (unsafe response).
> However, Eq. (16) (SafeDPO without $\Delta$) is insufficient to reflect the preference for (safe response) >> (unsafe response).
> Thus, we introduce $\Delta$ to address this issue.
> Although $\Delta$ is not needed for infinitely many samples, with finite samples, a larger $\Delta$ strongly encourages preferring safe responses (even if they are unhelpful) over unsafe responses (although they are helpful).
> Therefore, $\Delta$ balances the trade-off between helpfulness and safety when finite samples are used - this explains why safety increases while helpfulness decreases as $\Delta$ increases.
> In addition, with a very large $\Delta$, SafeDPO suffers from degeneration because SafeDPO with large $\Delta$ becomes similar to the unlikelihood method introduced in the DPO paper, as explained in the Appendix B.5.
>
> ### [Theoretical Analysis]
> **(Related to Weakness 2, 3)**
>
> First of all, $r(x,y)$ and $c(x,y)$ are unknown reward and cost functions, which are not accessible in practice.
> As noted in the reviewer’s comment, $r(x,y)$ in SafeDPO serves the same role as $r(x,y)$ in DPO.
> Since these functions are not accessible, they must be replaced by accessible terms, such as helpfulness preferences and safety indicators.
>
> However, such replacements can introduce bias, that is why we aim to prove that our replacements are theoretically unbiased.
> To address this, we introduce three propositions.
>
> Theoretically, if $r$ and $c$ were accessible, sampling preferences from $r\_c$ would suffice to solve the safety alignment objective implicitly.
> However, we cannot access these functions, we replace this sampling with helpfulness preference and safety indicators provided in the dataset.
> Proposition 3.2 demonstrates that the probabilities derived from sampling helpfulness preferences, safety indicators, and reordering preferences are equivalent to the probabilities obtained by sampling preferences from $r\_c$.
> In other words, informally, our replacement is theoretically equivalent to sampling directly from $r\_c$.
>
> ### [Efficiency of SafeDPO]
> **(Related to Weakness 4)**
>
> Please refer to our general response.
>
> ### [Hard to understand]
> **(Related to Weakness 5)**
>
> For now, we have added Figure 2 to help clarify the flow of the main section, and we will further improve readability in a future revised version.

---

> > ### Comment · Reviewer_WZ5n · 2024-11-21
> >
> > Thank you for your response. I have raised my rating accordingly.

---

> ### Author Response · Authors · 2024-11-25
>
> Thank you for taking the time to provide such thoughtful feedback.
> Your feedback is not only helpful but also motivates us to refine our study further.

---

### Official Review · Reviewer_jVNa · 2024-11-07

**Soundness:** 3
**Presentation:** 2
**Contribution:** 2
**Rating:** 6
**Confidence:** 4

**Summary:**

The paper introduces SafeDPO, an approach for enhancing the safety of LLMs by incorporating safety constraints into the process of reinforcement learning from human feedback (RLHF). The innovation of SafeDPO, compared to existing work like safe RLHF, is its ability to directly consider and optimise safety alignment without the need to train reward and cost models. Empirical evidence comparing SafeDPO against state-of-the-art approaches (Safe RLHF and three DPO variants) demonstrates its ability to minimise the probability of yielding unsafe responses while selecting the most preferable outputs between the set of feasible (safe) outputs.

**Strengths:**

+ SafeDPO is an interesting approach that aims to address the challenging problem of making LLMs produce outputs that are more closely aligned with human-based safety considerations. There is an increasing body of research investigating this problem, and as such, SafeDPO aims to contribute in this direction.

+ Good set of experiments using state-of-the-art approaches and building on the experimental setup utilised in existing research.

**Weaknesses:**

- The work, albeit interesting, appears incremental compared to Safe RLHF and the other SoTA research it is compared against, with the improvements being marginal. Please be explicit about the innovations compared to Safe RLHF and DAAs (Rafailov et al., 2024a).

- A key statement underpinning SafeDPO is that it is efficient in terms of computational time, memory usage, and data requirements. However, no evidence is provided to corroborate this statement. This is a major limitation, and the authors should either provide evidence (e.g., complexity analysis, efficiency improvement, training time, memory usage) compared to the baselines they compared their work against or revise their statements accordingly.

- The 'mildness' of the assumptions underpinning the core of SafeDPO (especially the statements underpinning Propositions 3.1 and 3.3) is not articulated convincingly, and the authors are encouraged to revisit these points.

**Questions:**

(Q1) It is reported that the intractable distribution is replaced by a tractable distribution without *theoretically* introducing any bias (Section 3.2). This is a key part of the theoretical framework underpinning SafeDPO. Could you please elaborate on the correctness of the above statement, as it is not entirely clear how this was achieved based on the information provided in this section?

(Q2) In several places throughout the paper (including the main contributions), you report that SafeDPO is efficient in computation time, memory usage, and data requirements. Unless I have missed this information in the main paper or the Appendix, I cannot see any evidence corroborating these claims. Could you please provide some details that would enable developing an understanding of the foundation for these improvements yielded by SafeDPO?

(Q3) Propositions 3.1 and 3.3 are founded on "mild assumptions". Although the proofs are provided in Appendices A.1 and A.3, could you please clarify the statement on the assumptions underpinning these propositions and provide a justification why these assumptions are considered mild?

(Q4) The impact of hyperparameter Δ in Equation (17) is briefly illustrated through the experiments presented in Section 5. Please elaborate on what this means from a practical perspective and how the Δ value could be calibrated or selected depending on the considered task.

**Details Of Ethics Concerns:**

Appendix B.1 reports an experiment carried out involving five participants. Consequently, it is understood that the appropriate protocols involving human subjects should have been realised for the execution of these experiments.

---

> ### Author Response · Authors · 2024-11-21
>
> Thank you for your constructive feedback and comments.
>
> ### [Contribution of SafeDPO]
> **(Related to Weakness 1)**
>
> Please refer to our general response.
>
> ### [Efficiency of SafeDPO]
> **(Related to Q2, Weakness 2)**
>
> Please refer to our general response.
>
> ### [Theoretical Analysis]
> First, we apologize for any unclear parts.
> We are currently revising the main section to improve clarity, but this process is time-consuming.
> During this rebuttal, we are primarily focusing on clarifying our explanation of the derivations.
>
> **(Related to Q1)**
>
> Theoretically, if $r$ and $c$ were accessible, sampling preferences from $r\_c$ would suffice to solve the Eq. (14).
> However, we cannot access these functions, we replace this sampling with helpfulness preference and safety indicators provided in the dataset.
> Proposition 3.2 demonstrates that the probabilities derived from sampling helpfulness preferences, safety indicators, and reordering preferences are equivalent to the probabilities obtained by sampling preferences from $r\_c$.
> In other words, informally, our replacement is theoretically equivalent to sampling directly from $r\_c$.
>
> **(Related to Q3, Weakness 3)**
>
> For theoretical derivations, we introduce two assumptions, Assumptions A.1. and A.2., as described in Appendix A.
> Based on this question, we have updated this section in the revised version.
>
> **Assumption A.1**: This assumption assumes that the reference model can generate at least one safe response with probability greater than $\epsilon$.
> Intuitively, this is essential because aligning a LM to generate safe answers requires the existence of at least one safe response.
> Furthermore, this is not a strong assumption since, in principle, we can always provide a safe but uninformative response, such as “we cannot answer this question”.
> While such a response may lack utility, it makes this assumption satisfied.
>
> **Assumption A.2**: This assumption simplifies the derivation by restricting the range of rewards.
> Without this assumption, the theory can still be derived by shifting and rescaling the reward function to map (x,y) pairs into the fixed range with high probability.
> However, adopting this assumption allows for a more concise and clear formulation of the derivation.
>
> ### [About $\Delta$]
> **(Related to Q4)**
>
> The core idea of SafeDPO is reordering: (safe and helpful response) > (safe but unhelpful response) >> (unsafe response).
> However, Eq. (16) (SafeDPO without $\Delta$) is insufficient to reflect the preference for (safe response) >> (unsafe response).
> Thus, we introduce $\Delta$ to address this issue.
> Although $\Delta$ is not needed for infinitely many samples, with finite samples, a larger $\Delta$ strongly encourages preferring safe responses (even if they are unhelpful) over unsafe responses (although they are helpful).
> Therefore, $\Delta$ trades-off between helpfulness and safety when finite samples are used.
>
> ### [Human Evaluation]
> **(Relate to Ethics Concern)**
>
> Participation is entirely voluntary, with individuals 18 or older asked to evaluate the safety and helpfulness of language model responses.
> While the task involves potentially sensitive or harmful content (e.g., curse words, violence, or adult material), participants are informed of the risks upfront and can opt out or skip questions at any time.
> Privacy is strictly maintained, and participants can withdraw without any penalty.
> We also inform participants that the purpose of this study is to enhance the safety of language models, and any concerns can be directed to the provided contact.

---

> > ### Author Response · Authors · 2024-11-25
> >
> > Dear Reviewer jVNa,
> >
> > As the rebuttal deadline approaches, we hope that our responses have satisfactorily addressed all of your concerns.
> > Should you need further clarification or additional explanations, we would be more than happy to provide them.
> >
> > Thank you for your time and consideration.
> >
> > Sincerely,
> >
> > The authors of SafeDPO.

---

### Author Response · Authors · 2024-11-21
**General Response to Reviewers**

We appreciate all the reviewers for their valuable feedback and thoughtful comments.
Below, we have summarized and addressed the several key points in this response.
If you have any further questions or concerns, please feel free to reach out to us.

## [Contribution of SafeDPO]
While RLHF achieves promising performance in preference alignment, it comes with high costs, particularly in hyperparameter optimization, computation time, and memory usage during training.
In contrast, DAAs offer a cost-effective alternative, achieving comparable performance to RLHF at significantly lower costs.
As the size of LLMs continues to grow and associated costs rise, the cost-efficiency of DAAs becomes an increasingly attractive and practical option.
Consequently, many recent studies have adopted DAAs for preference alignment in LLMs, with examples: Mixtral [1], Llama 3 [2], EXAONE 3.0 [3], Step-DPO [4], IterativeRPO [5], and many other methods.

In this context, some efforts have begun exploring DAA methods for safety alignment: C-DPO [6], and MoCAN and PeCAN [7].
To the best of our knowledge, SafeDPO is the first approach to achieve safety alignment without requiring additional models beyond the reference model ($\pi\_\text{ref}$) and current model ($\pi\_\theta$).

|                        | RLHF                          | DAA                 |
|------------------------|-------------------------------|---------------------|
| **Preference alignment** | RLHF (PPO)                    | DPO, IPO, SLiC-HF   |
| **Safety alignment**     | Safe RLHF (PPO-$\lambda$)     | SafeDPO             |

In addition, SafeDPO provides theoretical analysis that SafeDPO implicitly solves the safety alignment objective, i.e., If infinite data and perfect optimization are possible, the policy obtained through SafeDPO (Eq. (17) in our paper) and safety alignment (Eq. (8) in our paper) would be the same.

## [Efficiency of SafeDPO]
**Memory efficiency**
| Algorithm   | $\pi\_\text{ref}$ | $\pi\_\theta$ | Reward | Reward value | Cost | Cost value |
|-------------|---------------|--------------|--------|--------------|------|---------|
| Safe RLHF   | O                | O            | O      | O            | O    | O          |
| SafeDPO     | O                | O            | X      | X            | X    | X          |

Note that each network is a 7B parameter network. In addition, Safe RLHF suffers from out-of-memory with a machine with eight A100 40GB GPUs.

**Time efficiency (sec)**
| Algorithm   | Policy training  | Reward training | Cost training |
|----------|-------------|-----------|----------|
| SafeDPO     | 1388.2          | -               | -                |
| Safe RLHF   | 32957.1  | 1121.3    | 1121.9    |

Times for Safe DPO and Safe RLHF are measured using the implementations based on the public SafeRLHF code and a machine with sixteen A100 40GB GPUs (with 3 seeds). We omit the standard errors as they are $\le$1% of the corresponding values.

**Dataset efficiency**
| Algorithm   | Helpfulness preference | Safety indicator | Harmlessness preference |
|-------------|-------------------------|------------------|--------------------------|
| Safe RLHF   | O                       | O                | O                        |
| SafeDPO     | O                       | O                | X                        |

**Hyperparameters to search**
- Compared to DPO, SafeDPO uses only one additional hyperparameter, $\Delta$.
- Compared to DPO, Safe RLHF uses the following (see details in Appendix C of Safe RLHF paper [8]):
  - Hyperparameters for reward and cost models
      - epochs, regularization, lr, weight decay
  - Hyperparameters for PPO
    - critic lr, critic weight decay, critic lr schedular type, critic lr warmup ratio, ptx coeff $\gamma$, clip range ratio $\epsilon$
  - Hyperparameters for safety
    - threshold $-d$, lambda init $\lambda\_0$, lambda lr $\alpha$

The efficiency analysis of SafeDPO has been included in Appendix B.6 in the revised version.

## [Inconsistency in Helpful Evaluations]
GPT-4 tends to classify safer answers as more helpful, which suggests it may have been trained to favor safer responses.
To mitigate such bias, we additionally compare the helpfulness among the safe responses, as reported in Figure 3 of our paper.
In our revised paper, we have added abundant evaluations and examples in Appendix D to illustrate this bias.
In Appendix D.1, we have conducted further GPT-4 evaluation using the template from Appendix of C.2 of our paper, Appendix K of [7], and Appendix C.2 of [8].
In Appendix D.2, we have provided examples generated by DPO-HELPFUL and SafeDPO and their scores evaluated by these templates.
In the provided examples, harmful responses are evaluated as unhelpful responses by GPT-4, even if they directly answer the given questions, regardless of the templates used.
In Appendix D.3, we also evaluate the examples in Appendix N of [7] using two templates, used in this paper and the paper [7].

---

> ### Author Response · Authors · 2024-11-21
> **References**
>
> ### References
> [1] Jiang, Albert Q., et al. "Mixtral of experts." arXiv preprint arXiv:2401.04088 (2024).
>
> [2] Dubey, Abhimanyu, et al. "The llama 3 herd of models." arXiv preprint arXiv:2407.21783 (2024).
>
> [3] An, Soyoung, et al. "EXAONE 3.0 7.8 B Instruction Tuned Language Model." arXiv e-prints (2024): arXiv-2408.
>
> [4] Lai, Xin, et al. "Step-dpo: Step-wise preference optimization for long-chain reasoning of llms." arXiv preprint arXiv:2406.18629 (2024).
>
> [5] Pang, Richard Yuanzhe, et al. "Iterative reasoning preference optimization." arXiv preprint arXiv:2404.19733 (2024).
>
> [6] Liu, Zixuan, Xiaolin Sun, and Zizhan Zheng. "Enhancing llm safety via constrained direct preference optimization." arXiv preprint arXiv:2403.02475 (2024).
>
> [7] Huang, Xinmeng, et al. "One-Shot Safety Alignment for Large Language Models via Optimal Dualization." arXiv preprint arXiv:2405.19544 (2024).
>
> [8] Dai, Josef, et al. "Safe rlhf: Safe reinforcement learning from human feedback." arXiv preprint arXiv:2310.12773 (2023).

---

### Meta-Review · Area_Chair_Lh9Q · 2024-12-17

**Metareview:**

There are several technical errors that invalidate the main theoretical contribution of this work. The reviewers seem to have missed this; however, the authors claim that a Galerkin relaxation of an almost sure constraint (equation 8) is equivalent to its relaxation in equation (9) in Proposition 4.1 "under mild assumptions" -- however, these assumptions are never specified in the appendix, and are unlikely to reasonably hold for any class of constraints. Therefore, the core methodology is based on faulty logic, and the central claims are unproven.

**Additional Comments On Reviewer Discussion:**

The reviewers seemed to have mostly focused on the experimental analysis of the proposed technique, which indeed is promising. But without any methodologically sound contribution underlying them, I cannot justify the publication of this work.

---

> ### Public Comment · ~Moontae_Lee1 · 2025-02-14
>
> After carefully reviewing the meta-review, there may have been a significant misunderstanding that requires scientific clarification.
>
> 1. The draft has never claimed that Equation (8) is equivalent to Equation (9). Unfortunately, this false assumption appears to have been the primary basis for the rejection.
> * The authors seem to believe that these two equations are not equivalent.
> 2. In fact, the objective is explicitly derived from Equation (8).
> * As substantiated by Assumptions A.1 and A.2, as well as Propositions 4.1, 4.2, and 4.3.
>
> Given the positive evaluations from all 5 reviewers who carefully assessed both the theoretical and empirical contributions, it is concerned that the meta-review may have disproportionately relied on a misinterpretation of our theoretical framework. While  fully respecting the meta-reviewer’s right to adopt a critical stance, when overriding multiple reviewers’ positive assessments, it is essential that any critiques are grounded in clear and scientifically verifiable evidence.

---

### Decision · Program_Chairs · 2025-01-22

Reject

---

> ### Public Comment · ~Moontae_Lee1 · 2025-02-14
>
> While fully appreciating ICLR's commitment to maintaining a thorough review process, it is necessary to express significant concerns regarding the meta-review of our manuscript.
>
> The meta-review identifies our work as flawed based on an incorrect assumption—one that our submission never claimed. If there are indeed issues with the logical flow of our arguments, we would be grateful for the opportunity to address them in detail. Ensuring that the review accurately reflects the content and contributions of the submission is essential for supporting the high standards that ICLR represents. This principle should apply to the meta-review as well.